# Palladium-Catalyzed Organic Reactions Involving Hypervalent Iodine Reagents

**DOI:** 10.3390/molecules27123900

**Published:** 2022-06-17

**Authors:** Samata E. Shetgaonkar, Ritu Mamgain, Kotaro Kikushima, Toshifumi Dohi, Fateh V. Singh

**Affiliations:** 1Chemistry Division, School of Advanced Sciences (SAS), Vellore Institute of Technoloy-Chennai, Chennai 600127, Tamil Nadu, India; samatashetgaonkar@gmail.com (S.E.S.); ritu.mamgain@vit.ac.in (R.M.); 2College of Pharmaceutical Sciences, Ritsumeikan University, 1-1-1 Nojihigashi, Kusatsu 525-0058, Shiga, Japan; kixy@okayama-u.ac.jp

**Keywords:** palladium, hypervalent iodine reagents, oxidant, catalyst, bond formation

## Abstract

The chemistry of polyvalent iodine compounds has piqued the interest of researchers due to their role as important and flexible reagents in synthetic organic chemistry, resulting in a broad variety of useful organic molecules. These chemicals have potential uses in various functionalization procedures due to their non-toxic and environmentally friendly properties. As they are also strong electrophiles and potent oxidizing agents, the use of hypervalent iodine reagents in palladium-catalyzed transformations has received a lot of attention in recent years. Extensive research has been conducted on the subject of C—H bond functionalization by Pd catalysis with hypervalent iodine reagents as oxidants. Furthermore, the iodine(III) reagent is now often used as an arylating agent in Pd-catalyzed C—H arylation or Heck-type cross-coupling processes. In this article, the recent advances in palladium-catalyzed oxidative cross-coupling reactions employing hypervalent iodine reagents are reviewed in detail.

## 1. Introduction

Polyvalent iodine compounds have become popular in organic synthesis as they have been proven to be efficient and eco-friendly reagents [1,2,3]. Furthermore, these reagents are non-toxic, quite stable, simple to prepare, and a viable alternative to metal-derived oxidants or catalysts in a variety of oxidative reactions [4,5,6,7,8,9]. Various scientific papers, book chapters, and review articles have been published on the chemistry of hypervalent iodine. [10,11,12,13,14,15,16,17,18]. The functionalization of carbonyl compounds [19,20], cyclization [21,22,23,24], oxidative rearrangements [25,26,27], alkene difunctionalizations [28,29,30,31], and atom-transfer reactions [32], in the presence of hypervalent iodine compounds as reagents or catalysts, is now well established. The capacity of hypervalent iodine reagents to operate as both oxidant and ligand transfer reagents is the key to the substantial success made in this area [33,34].

Palladium, on the other hand, has emerged as a versatile catalyst. It is an essential component of several coupling reactions, such as Stille coupling and the Suzuki–Miyaura, Heck, Buchwald–Hartwig, Sonogashira, and Negishi, resulting in a broad variety of useful compounds [35]. The effect of hypervalent iodine in palladium-catalyzed reactions has received a great deal of attention over the years. In 2007, Sanford and colleagues published the first review paper addressing the unusual reactivity of hypervalent iodine reagents in Pd-catalyzed reactions [36]. Wengryniuk’s group later published a piece of a review in 2017 that outlines the critical significance of polyvalent iodine reagents in high-valent palladium chemistry [37]. Polyvalent iodine compounds react efficiently with palladium complexes due to their electrophilic nature and oxidizing property, promoting reactions through Pd(0/II) and Pd(II/IV) catalytic cycles [38]. Furthermore, a handful of these se compounds are used as aryl, alkynyl, and heteroatom ligand sources in several Pd-catalyzed ligand transfer processes. The commonly used hypervalent iodine(III)/(V) reagents in palladium-catalyzed reactions are listed in Figure 1. Hypervalent iodine(III)/(V) reagents, such as phenyliodine(III) diacetate **1** (PIDA), phenyliodine(III) bis(trifluoroacetate) **2** (PIFA), phenyliodine(III) dipivaloate **3** (PIDP), and Dess-Martin periodinane **4** (DMP), are frequently employed oxidants in palladium-catalyzed reactions. Apart from this, cyclic hypervalent iodine(III) reagents **5** and **6** are also used as oxidants, whereas 1-[(triisopropylsilyl)ethynyl]-1,2-benziodoxol-3(1*H*)-one **7** (TIPS-EBX) is widely explored as an alkylating reagent. Owing to their highly electrophilic nature, diaryliodonium salts **8** are excellent arylating reagents in palladium-catalyzed reactions.

In comparison to the Pd(0/II) redox cycle, substantial progress has been achieved in the chemistry of Pd(II/IV)-catalyzed reactions over the last several decades. In this context, the current review focuses on recent progress in palladium-catalyzed transformations utilizing hypervalent iodine reagents, emphasizing possible synthetic applications and mechanistic features. The article is categorized based on the bonds generated, which include C—O, C—N, C—C, C—Si, C—B, and C—halogen bonds, as well as alkene difunctionalization.

## 2. C—O Bond Formation

Palladium-catalyzed, ligand-mediated C—H functionalization has been known to be one of the most effectual, atom-efficient, and cost-effective methods for introducing different functional groups to unactivated arene and alkane C—H bonds in organic synthesis. Several research groups have conducted extensive studies on the formation of the C—O bond, using palladium catalysis involving hypervalent iodine reagents as the oxidant or heteroatom ligand. Pd-catalyzed C—H oxidative cyclization, C—H acyloxylation, C—H alkoxylation, and allylic oxidation are significant methods in C—O bond formation that are guided by directing functional groups such as oxime ether, oxazoline, amide, pyridine, pyrimidine, and so on.

### 2.1. C—H Cyclization

Significant progress has been made in the field of palladium-catalyzed oxidative cyclization processes employing hypervalent iodine reagents, which allow access to a variety of oxygen-containing heterocycles. For instance, Yu and co-authors developed a novel method for the construction of dihydrobenzofurans **10** via the palladium-catalyzed C—H activation/C—O cyclization reaction [39]. In the presence of (diacetoxyiodo)benzene **1** as the terminal oxidant and Pd(OAc)_2_ as the catalyst, a series of tertiary alcohols **9** were efficiently converted into targeted cyclized products **10** in moderate to good yields (Figure 1). Moreover, the scope of the reaction was extended for the preparation of important scaffolds such as spirocyclic dihydrobenzofurans.

The catalytic cycle for the preparation of dihydrobenzofurans **10** was initiated by the palladium-catalyzed C—H activation of substrate **9** to give intermediate **11,** followed by subsequent oxidation using PhI(OAc)_2_ **1** to give Pd(IV) intermediate **12**. Finally, the reductive elimination of **12** gives cyclized product **10** along with the regeneration of the Pd(II) catalytic species to continue the catalytic cycle (Figure 2).

Later, Gevorgyan and coworkers achieved the intramolecular silanol group-directed C—H oxygenation of arenes **13** using PIDA **1** as an oxidant in the presence of a palladium catalyst [40]. These reactions begin with the production of cyclic silicon-protected catechols **14**, which are then desilylated with TBAF/THF to yield substituted catechols **15** (Figure 3). The reaction featured excellent site selectivity and broad substrate scope, particularly as electron-rich substrates react much faster and provide high yields. 

The probable catalytic cycle begins with the coordination of Pd with silanols **13** to generate palladacycle **16**, followed by PIDA-mediated oxidation to produce Pd(IV)-intermediate **17**. The intermediate **17** is then reductively acetoxylated into intermediate **19** and regenerates the palladium catalyst. Finally, acid-catalyzed transesterification of **19** yields **21** and loses acetic acid to generate cyclic silyl-protected catechols **14**, which are then desilylated with Tetrabutylammonium fluoride (TBAF) **18** to yield catechols **15**. Based on ^18^O-labeling experiments, the production of product **14** by direct C—O reductive cyclization was ruled out (Figure 4). 

Another interesting work published by Gevorgyan’s research group is a convenient method for the synthesis of oxasilacycles **23** and **25** from benzyl-silanol **22** and **24**, respectively, using a combination of Pd(OAc)_2_ and PhI(OAc)_2_ **1** via C—H oxygenation strategy [41]. Under the optimized conditions, a variety of silanol-directed aromatic substrates **22** and **24** bearing alkyl and aryl substituents were transformed into the corresponding cyclic products in significant yields (Figure 5). Gratifyingly, the oxasilacycles were found to be valuable intermediates as they contained an easily removable or modifiable Si—O bond and thus could be converted into useful functionality. The reactions include the well-known Tamao oxidation, Hiyama–Denmark cross-coupling, and nucleophilic addition, as well as the novel Meerwein salt-mediated oxasilacycle ring-opening and nitrone synthesis from the benzylsilane and nitroso compound. The desilylation of the cyclic product in the presence of CsF in DMF to give phenol in good yield is an example of the synthetic usefulness of oxasilacycles. 

Furthermore, Dong and colleagues reported the production of cyclic ethers **27** by palladium-catalyzed oxime-masked-alcohol-directed dehydrogenative annulation of the substrates **26** sp^3^ C—H bonds using (diacetoxyiodo)benzene **1** as an oxidant [42]. Under normal circumstances, the reaction proceeds preferentially at the β position, and the substrates **26** with the primary, secondary, and tertiary hydroxyl groups perform extremely well (Figure 6). The process might continue through C—H palladation, followed by Pd oxidation, to a higher oxidation state and an intramolecular S_N_2 reaction to generate oxonium intermediate **29**. Finally, cyclic ethers **27** were synthesized through deprotonation or debenzylation and used to renew the Pd catalyst.

Shi and colleagues demonstrated the Pd-catalyzed intramolecular lactonization of, α, α-disubstituted arylacetic acids **30** in the presence of PhI(OAc)_2_ **1** and Ac—Gly—OH as the required ligand to obtain a variety of, α,α-disubstituted benzofuran-2-ones **31** in varying yields [43]. The catalytic system is made up of Pd(OAc)_2_ and a mixture of NaOAc, CsOAc, and AgOAc as the most efficacious bases (Figure 7). Wang et al., in 2013, proposed a similar C—H activation/C—O production technique for constructing functionalized benzofuranones [44].

The proposed mechanistic approach for the lactonization of acids **30** is outlined in Figure 8. The reaction begins with the deprotonation of acid **30** under the basic condition to form carboxylate salt **32**, which further coordinates with the Pd catalyst to give intermediate **33**, followed by C—H cleavage via concerted metalation deprotonation to form Pd(II) intermediate **34**. Six-membered palladacycle **34** underwent oxidation with PhI(OAc)_2_ **1** to give Pd(IV) intermediate **35**, which underwent final reductive elimination to release product **31** via path ‘a’ or formed acetoxylated product **36**, which then condensed to give anticipated product **31** (path ‘b’) 

Subsequently, a novel route to construct biaryl lactones **38** from biaryl carboxylic acids **37** via palladium-catalyzed C—H activation/C—O cyclization, using PhI(OAc)_2_ **1** as an effective oxidant, was developed [45]. The presence of acetyl-protected glycine (15 mol% Ac—Gly—OH) as a ligand, along with base KOAc and solvent t-BuOH, provided the best results for the desired products **38** (Figure 9). Both the electron-rich and the electron-deficient substituents were well tolerated on the aryl rings. Furthermore, the present protocol was successfully utilized for the total synthesis of the natural product cannabinol in a 72% yield.

Shi’s group, in 2016, revealed a straightforward method for the synthesis of γ-lactones **40** via the Pd(II)-catalyzed 2-pyridinylisopropyl (PIP) auxiliary-directed intramolecular cyclization of unactivated C(sp^3^)—H bonds, utilizing the oxidant PIDA **1** [46]. The lactonization of aliphatic acids **39** with different substituents on the alkyl chain went exceptionally well, yielding γ-lactones **40** in 32—77% yields (Figure 10).

The formation of a five-membered palladacycle **41** via Pd-catalyzed C—H activation facilitated by bidentate auxiliary is the most plausible pathway for the lactonization of aliphatic acids **39**. In the presence of PhI(OAc)_2_ **1**, palladacycle **41** was oxidized to provide Pd(IV) intermediate **42**, which was then ligand exchanged to generate **43** and was further reductively eliminated to liberate target product **40** and a Pd(II) catalyst to sustain the catalytic cycle (Figure 11). Another route to lactone **40** is by a direct S_N_2-type attack by the carboxylate group on the Pd(IV)—C bond of **42**.

### 2.2. C(sp^2^/sp^3^)—H Acyloxylation

Over the years, C—H acyloxylation has gained considerable attention because it introduces ester functionality on the aromatic and aliphatic substrates. Using palladium catalysts and iodine(III) reagents as oxidants, notable progress has been achieved in transmuting sp^2^ and sp^3^ hybridized C—H bonds into useful C—O bonds. In the next section, we will discuss the recent developments made in C—H acyloxylation reactions employing different directing groups.

#### 2.2.1. C(sp^2^)—H Acyloxylation

Several ligand-directed C(sp^2^)—H acyloxylation reactions have been developed, giving facile access to valuable oxygenated arenes. In 2009, Chen and co-authors published the Pd(II)-catalyzed pyrimidine-directed ortho-acetoxylation of phenol derivatives **44** in the presence of PhI(OAc)_2_ **1** as an efficient oxidant in combination with the Ac_2_O/AcOH solvent system [47]. The reaction proceeded through the Pd-catalyzed ortho C—H activation of pyrimidyl ethers **44**, resulting in the formation of six-membered palladacyles which, upon functionalization, furnished acetoxylated products **45** in variable yields (Figure 12). However, the substrates **44** with electron-withdrawing groups or with ortho-/meta-substituents reacted slowly and gave the desired products in moderate yields.

Later, Liang and his co-workers employed the bidentate ligand system for the Pd(II)-catalyzed C—H activation/C—H acetoxylation of amide substrates **46** and **48** [48]. Under the optimized conditions, various pyridines **46** and 8-aminoquinoline **48** derivatives were converted to the desired acetoxylated products **47** and **49**, respectively, in the presence of PhI(OAc)_2_ **1** as an oxidant as well as an acetate source (Figure 13). 

The plausible catalytic cycle for the ortho-acetoxylation of arenes is outlined in Figure 14. The reaction begins with the coordination of amide substrates **46** with a Pd(II) catalyst to give 5-membered fused palladacyles **50**, followed by oxidation with PhI(OAc)_2_ **1** to form an unstable Pd(IV) intermediate **51**, which subsequently undergoes reductive elimination to give acetoxylated products **47**.

In 2010, Sanford and co-author employed in situ-generated O-acetyl oxime as an efficient directing group for the sp^2^ C—H acetoxylation of **52 [49]**. The reaction involves the O-acetylation of oximes **52**, occurring upon treatment with AcOH/Ac_2_O for 2 h at 25 °C, to form O-acetylated products **53**, which further direct C—H acetoxylation in the presence of Pd(OAc)_2_ and PhI(OAc)_2_ **1** to afford mono-ortho-oxygenation products **54** (Figure 15). Furthermore, the synthesized compounds were readily transformed into valuable compounds such as ketones, amines, alcohols, and heterocycles using different reaction conditions. 

Later, Gevorgyan and co-workers described the pyridyldiisopropylsilyl (PyDipSi)-directed C—H acetoxylation/pivaloxylation of arenes through palladium catalysis [50]. Arylsilanes **55** reacts in the presence of hypervalent iodine(III) reagents PhI(OAc)_2_ **1** or PhI(OPiv)_2_ **56** in 1,2-dichloroethane (DCE) to yield monoacetoxylated or pivaloxylated products **57** in a good yield (Figure 16). Both of the hypervalent iodine(III) reagents act as oxidants as well as the source of the acyloxyl group. The reaction possessed an easily removable directing group, and it possessed remarkable functional group tolerance and excellent site selectivity. 

Furthermore, the same group performed double C—H pivaloxylation of the 2-pyrimidyldiisopropylsily (PyrDipSi)-directed arenes **58** to afford bispivaloxylated products **59**, using the Pd(OAc)_2_/PhI(OPiv)_2_ **56** catalytic system [51]. Additionally, the ortho-substituted arenes **60** smoothly transformed into monopivaloxylated products **61** in good yields under similar conditions (Figure 17). Finally, the PyrDipSi group was easily removed to yield protected resorcinols or was converted into useful synthetic products.

In 2013, Shi and co-workers employed 1,2,3-triazoles-pyridine (TA-Py) as a directing group in the Pd(II)-catalyzed selective ortho-C—H activation of arenes **62** for the first time, using oxidant PhI(OAc)_2_ **1** and co-oxidant AgOAc [52]. The reaction scope was examined with various TA-Py amides **62** to furnish the desired oxidized products **63** in useful yields (Figure 18). In the case of the meta-substituted arenes, excellent regioselectivity (dr > 20:1) was achieved with acetoxylation, taking selectively at less sterically hindered carbon. A further TA-Py group also promoted the acetoxylation of unactivated sp^3^ C—H substrates under identical conditions.

In 2015, Dong’s team published the Pd-catalyzed dimethoxybenzaldoxime-directed ortho-acetoxylation of arenes **64**, using oxidant PIDA **1** [53]. Both the primary and the secondary masked alcohol-derived substrates **64** smoothly underwent ortho-acetoxylation to yield acetoxylated products **65** in good to excellent yields (Figure 19). The substrates **64** with ortho- or meta-substituents gave mono-oxidation products, while the symmetrical substrates formed bis-oxidation products.

In addition, a regioselective approach, including the Pd(II)-catalyzed C—H benzoxylation of 2-arylpyridines **66**, yielded mono-benzoxylation products **68** in moderate to good yields [54]. They used the easily accessible iodobenzene dibenzoate derivatives **67** as an oxidant and benzoxyl group source (Figure 20). Furthermore, the current benzoxylation process was effectively employed for the benzoxylation of 2-thienyl pyridines **69** to obtain 3-benzoxylated thiophenes **70** in a high yield.

Figure 21 depicts a probable mechanism for the ortho C—H benzoxylation process. Initially, the substrates **66** are activated using a palladium catalyst to generate complex **71**, which is then oxidatively added to **67** to form complex **72** in a high oxidation state Pd(IV) or a Pd(III)-Pd(III) intermediate [55]. Finally, the reductive elimination of **72** yields the desired product **68** and regenerates the palladium catalyst, bringing the catalytic cycle to a close.

The Pd-catalyzed C—H oxygenation of simple arenes devoid of directing groups remains a challenge as it leads to the formation of mixtures of isomers. Sanford and colleagues discovered the nonchelated-aided Pd-catalyzed C—H acetoxylation of simple arenes, utilizing pyridine as a ligand [56]. Later, the same research group studied the use of the oxidant and ligand in controlling the site selectivity in the Pd-catalyzed C—H acetoxylation of multi-substituted arenes **73** [57] (Figure 22). Under ligand-free conditions and in the presence of PhI(OAc)_2_ **1**, the C—H acetoxylation of arenes **73** gave a modest yield of products **75** with the selectivity dominated by electronic effects, resulting in preferential acetoxylation at the electron-rich sites (Conditions A). On the other hand, the use of acridine (1.5 mol%) as an ancillary ligand in combination with that of Pd(OAc)_2_ and MesI(OAc)_2_ **74** showed sterically controlled selectivity (Conditions B).

Furthermore, the regioselective C—H functionalization of indoles has been discovered to be a simple approach for obtaining physiologically relevant 3-acetoxyindoles. Suna’s and Kwong’s groups both separately reported the synthesis of 3-acetoxyindoles **77** by the Pd(II)-catalyzed direct C3-oxidation of indole derivatives, employing PhI(OAc)_2_ **1** as an efficient terminal oxidant [58,59]. In addition, Lei and colleagues used PhI(OAc)_2_ **1** and KOH as bases to establish a comparable Pd-catalyzed method for the selective C3-acetoxylation of substituted indoles **76** [60]. Mechanistic studies indicated that electrophilic palladation occurs at the C3 position of indole to form a Pd(II) species, which is oxidized to a Pd(IV) intermediate and then reductively eliminated to provide matching C3-acetoxylated indoles **77**. (Figure 23).

Szabó and co-workers presented an excellent example for the preparation of allylic acetates or benzoates **80** via the Pd-catalyzed allylic C—H acetoxylation/benzoyloxylation of alkenes **78**, using hypervalent iodine reagent as an oxidant [61]. The reactions were carried out in AcOH or MeCN solvent in the presence of bases KOAc and LiOBz. The catalytic process involved the formation of (η^3^-allyl)palladium intermediate **79**, which was confirmed through deuterium-labelling studies. Moreover, the reaction worked perfectly well for both the internal and the terminal alkenes to provide an exclusively trans product. (Figure 24). 

Later, the same group described the conversion of functionalized cyclic **81** or acyclic alkenes **84** into allylic trifluoroacetates **83** or **85** via Pd-catalyzed C—H trifluoroacetoxylation, employing PhI(OCOCF_3_)_2_ **82** as the oxidant and trifluoroacetoxy source [62]. Excellent regioselectivity (d.r: > 95:5) and diastereoselectivity were observed in the case of the monosubstituted cycloalkanes. Furthermore, the cyclic alkenes reacted much faster than the acyclic ones, and therefore, the addition of LiOCOCF_3_ was necessary in the case of substrates **84** (Figure 25). 

#### 2.2.2. C(sp^3^)—H Acyloxylation 

Another major approach in regioselective C—O bond formation is the acyloxylation of aliphatic C(sp^3^)—H bonds. Simple methods for activating a suitable C—H bond have been designed, utilizing various directing groups. In 2010, a new chelation-assisted Pd(OAc)_2_-catalyzed C(sp^3^)—H acyloxylation of 8-methylquinoline **86** was established in the presence of a stoichiometric amount of the oxidant PhI(OAc)_2_ **1** [63]. The reaction scope was investigated using a wide variety of carboxylic acids **87** to obtain mono-acyloxylation products **88** in moderate to good yields (Figure 26).

The authors proposed a mechanism for this C(sp^3^)—H acyloxylation reaction, which is outlined in Figure 27. The reaction began with the chelation-assisted activation of the benzylic C—H bond of **86** to form a cyclopalladated intermediate **89**, which underwent further oxidation with PhI(OCOR)_2_ **90** (formed in situ by the reaction of PhI(OAc)_2_ **1** and RCOOH **87** to form Pd(IV) intermediate **91**). Finally, the reductive elimination of **91** gave the desired product **88**. 

In 2010, Neufeldt and Sanford also reported the Pd-catalyzed in situ-generated O-acetyl oxime-directed sp^3^ C—H acetoxylation of dialkyl oximes **92** to afford acetoxylated products **94** in useful yields [49]. The acetoxylation reaction was compatible with different functional groups, such as alkyl chlorides, protected amines, and benzylic C—H bonds (Figure 28). Moreover, the acetoxylation occurs selectively at primary β sp^3^ C—H bonds, in comparison to the analogous secondary sites.

Using oxime as the directing group, the acetoxylation of β C—H bond of substrates **95** was performed, employing Pd(OAc)_2_ and PhI(OAc)_2_ **1** [64]. The catalytic reaction was expected to generate a five-membered exo-palladacycle **96**, which on oxidation gives masked 1,2-diols **97** (Figure 29). Moreover, the selective functionalization of the β-methylene (CH_2_) and β-methine (CH) groups in cyclic substrates was also carried out under the same reaction conditions. Furthermore, the deprotection of the DG and acetyl groups was conducted by using Zn/AcOH and K_2_CO_3_/MeOH, respectively, to yield diols in excellent yields.

An elegant protocol employing S-methyl-S-2-pyridyl-sulfoximine (MPyS) as the directing group for the selective catalytic oxidation of the unactivated primary β-C(sp^3^)—H bond of the amide substrates **98** at room temperature was developed [65]. In the presence of Pd(OAc)_2_ and PhI(OAc)_2_ **1**, the preparation of β C—H acetoxylated products **100** was achieved using carboxylic acids **87** as the solvent and acetate source (Figure 30). Furthermore, the diacetoxylation of the β, β^’^-C(sp^3^)—H bonds of amides **99** was also investigated under the modified conditions to afford diacteoxylated products **101**. 

Later, the benzylic C(sp^3^)—H bonds of **102** were subjected to Pd-catalyzed acetoxylation, using picolinamide and quinoline-2-carboxamide as efficient directing groups in the presence of PhI(OAc)_2_ **1** as the oxidant and acetate source [66]. This oxidative transformation furnishes acetoxylated products **103** with excellent functional group compatibility and broad substrate scope (Figure 31). Furthermore, the amide auxiliary was removed through base hydrolysis to give 2-aminobenzyl alcohols in a high yield. 

The proposed mechanism for this reaction is depicted in Figure 32. The reaction was initiated by the coordination of **102** with the Pd(II) catalyst to form palladacycle intermediate **104** via directed C—H activation, followed by oxidation in the presence of PhI(OAc)_2_ **1** and Ac_2_O to form Pd(IV) intermediate **105**. Finally, the reductive elimination of **105** gave the desired products **103** and the regenerated Pd catalyst to continue the catalytic cycle. 

In 2014, Chen and colleagues accomplished the Pd(OAc)_2_-catalyzed acetoxylation of the C(sp^3^)—H bond of simple alkylamines **106** guided by picolinamide (PA), utilizing PhI(OAc)_2_ **1** as an oxidant and under an argon atmosphere [67]. The procedure makes it simple to obtain acetoxylated compounds **107** in a high yield. Furthermore, under these conditions, the C—H acetoxylation of the methyl group of arylamines **108** progressed easily, yielding acetoxylated compounds **109**. The addition of Li_2_CO_3_ was crucial as it suppressed the formation of cyclic azetidine through intramolecular C—H amination (Figure 33).

Stambuli and co-workers reported a Pd-catalyzed PhI(OAc)_2_-mediated allylic oxidation of cis-vinylsilanes **110**, using PIDA **1** to give the corresponding cis-silyl allylic acetate **111** as the major product [68]. This ligand-free approach required lower catalyst loading and exhibited good substrate scope, and the oxidation products were isolated in moderate to good yields (Figure 34). 

Recently, in 2021, Punji and coworkers reported the palladium-catalyzed chemoselective C(sp^2^)—H and C(sp^3^)—H acetoxylation of tertiary amides through coordinated O—chelation under mild conditions [69]. On screening the reaction parameters, the best results were found on reacting substituted tertiary amide **112** with diacetoxyiodobenzene **1** (3.0 equiv.) in the presence of 1 mol% Pd(OAc)_2_ as a catalyst, dissolved in hexafluoroisopropanol (HFIP)/Ac_2_O at 80 °C for 20 h (Figure 35). On performing the reaction in acetic acid at 120 °C, the mono-acetoxylated product **113**, along with the diacetoxylated product, was obtained, but on reducing the temperature to 80 °C and performing the reaction in HFIP/Ac_2_O, a high selectivity of monoacetoxylation was observed. The mild inorganic oxidants, such as Na_2_S_2_O_8_, K_2_S_2_O_8_, and AgOAc, were found to be less effective in comparison to the PhI(OAc)_2_. The amides with cyclic substituents, as well as simple dialkyl amides with different steric properties, were found to be well-tolerated and yielded the desired acetoxylated compounds in good to excellent yields. Under the optimized conditions, the acetoxylation of the methylene C(sp^3^)—H bond on the tertiary and cyclic amides failed to occur. Similarly, simple carboxylic acid and ester could not afford the acetoxylated products. 

The possible catalytic cycle for the acetoxylation of tertiary amide is given in Figure 36. The reaction is assumed to start with the coordination of tertiary amide **112** to the Pd(II) species via carbonyl oxygen, which is followed by C—H cleavage, resulting in an alkyl-Pd(II) intermediate **114**. Furthermore, the intermediate **115** was obtained as the outcome of Pd(II) to Pd (IV) oxidation by PhI(OAc)_2_ **1**. Finally, the reductive elimination step results in the formation of the product as well as the regeneration of the Pd(II) catalyst for the next cycle. 

Ariafard and co-workers recently described the mechanism of the Pd(OAc)_2_-catalyzed alkoxylation of butyramide derivatives facilitated by hypervalent iodine(III) reagents, with the help of density functional theory (DFT) calculations. The calculations led to the result that the process consists of four basic steps: (i) C(sp^3^)—H bond activation, (ii) oxidative addition, (iii) reductive elimination, and (iv) active catalyst regeneration. The first step completes through a concerted metalation–deprotonation (CMD) mechanism. Furthermore, the oxidative addition begins with the transfer of an X ligand from a hypervalent iodine reagent (ArIX2) to Pd(II) to create a square pyramidal complex with an iodonium at the apical position. The Pd(II) oxidation is triggered by a straightforward isomerization of the consequent five-coordinate complex. As a result, moving the ligand trans to the Pd—C(sp^3^) bond to the apical position enhances the electron transfer from Pd(II) to iodine(III). This leads to the iodine(III) reduction accompanied by the release of the second ligand as a free anion. The C—O reductive elimination of the generated Pd(IV) complex is accomplished by the nucleophilic attack of the solvent (alcohol) on the sp^3^ carbon through an outersphere S_N_2 mechanism aided by the X anion. The oxidative addition and reductive elimination activities occur with a relatively low activation barrier (DG^‡^ 0–6 kcal mol^−1^). Due to the coordination between the alkoxylated product and the Pd(II) center, the regeneration of the active catalyst is endergonic. Thus, the subsequent catalytic cycles proceed with a substantially greater activation barrier in comparison to the initial catalytic cycle [70].

### 2.3. C(sp^2^/sp^3^)—H Alkoxylation

Another intriguing Pd-catalyzed reaction that allows the synthesis of C—O bonds is the -CH alkoxylation of sp^2^ and sp^3^ bonds, utilizing hypervalent iodine reagents as an oxidant. In 2012, Chen’s group reported a Pd(OAc)_2_/PhI(OAc)_2_ catalytic method featuring the alkoxylation of the C(sp^3^)—H bonds of picolinamide-coupled amines **116** at γ or δ positions, employing alcohols **118** as a source of the alkoxy group [71]. A series of alkyl ether products **119** were isolated in 42–95% yields with excellent functional group tolerance (Figure 37). In addition, the C(sp^2^)—H alkoxylation of arenes **117** was also investigated to yield mono- or bisalkoxylated products **120** in variable yields. Finally, the picolinamide auxiliary could be easily removed by treatment with an aqueous HCl/MeOH solution.

In 2013, Shi and co-workers developed a classic Pd-catalyzed method, in which a pyridine-based bidentate auxiliary enabled C—H alkoxylation of the methylene and methyl C(sp^3^)—H bonds of **121**, employing different alcohols **118** as the source of the alkoxy group mediated by PhI(OAc)_2_ **1 [72]**. Moreover, this catalytic oxygenation route was successfully applied to the alkoxylation of the β and γ C(sp^2^)—H bonds of arenes **123**. A facile synthesis of alkyl ethers **122** and aryl alkyl ethers **124** was achieved with good yields (Figure 38). Additionally, the directing group can be removed via nitrosylation and hydrolysis to yield β-methoxycarboxylic acid, which can be used for further transformation to various functional groups. 

Later, Sun et al. reported the azo group-directed selective C(sp^2^)—H alkoxylation of azobenzene compounds **125** with alcohols **118**, utilizing palladium catalysis [73]. Using this method, the synthesis of ortho-alkoxy aromatic azo scaffolds **126** was prepared with moderate to good yields, using both primary and secondary alcohol **118** as the alkoxylation reagents (Figure 39). However, alkoxylation occurred only with meta-substituted azobenzenes, while the ortho- and para-substituted azobenzenes gave the desired products in traces.

The proposed catalytic pathway for the o-alkoxylation of azobenzene derivatives **125** is depicted in Figure 40. Initially, the substrates **125** coordinate with the Pd catalyst which results into the formation of palladacyle **127** via C—H activation. Next, the PIDA-induced oxidation in the presence of alcohols **118** gave Pd(IV) intermediate **128**, which underwent reductive elimination to afford targeted products **126** and regenerated the palladium catalyst. 

Subsequently, Rao and his co-workers reported the first example of employing cyclic hypervalent iodine(III) reagents **130** as efficient oxidants in the Pd-catalyzed C(sp^3^)—H bond alkoxylation of unactivated methylene and methyl groups [74]. A series of 8-aminoquinoline-derived carboxylic acid substrates **129** and **132** were converted into the β-alkoxylated products **131** or **133**, respectively, by utilizing a variety of alcohols **118** as an alkoxy source (Figure 41). Furthermore, the synthetic application of the current approach for the alkoxylation of several Ibuprofen analogues, such as Naproxen, Ketoprofen, and Flurbiprofen, to obtain alkoxylated compounds in varying yields, was proven.

Later, the same research group established a similar approach for producing symmetrical acetals **137** through the Pd-catalyzed regioselective double C(sp^3^)—H bond alkoxylation of 8-aminoquinoline-derived substrates **125** with the alcohols **118**, using cyclic iodine(III) reagent **130** as an oxidant [75]. However, in the case of unsymmetrical acetals, as per the previous condition, the premixture of both alcohols **118** and **136** would give symmetric acetal as the major product (Figure 42). Therefore, a modified two-step protocol was developed wherein, initially, the monoalkoxylation of **134** with ROH **118** was carried out at 80 °C for 2–6 h, followed by the addition of R^2^OH **136** and the oxidant **130** to yield unsymmetric acetals **138** in good yields.

In 2014, Zhang and Sun demonstrated the regioselective alkoxylation ortho C(sp^2^)—H bond of 2-aryloxypyridines **139** to provide ortho-alkoxylation products **140** in the presence of a catalytic amount of Pd(OAc)_2_ and oxidant PhI(OAc)_2_ **1** [76]. The reaction employed 2-pyridyloxyl as an easily transformable directing group and alcohols **118** as a source of the alkoxy group (Figure 43). Electron-rich substrate-bearing groups, such as alkoxy and methyl, gave the best results, whereas the electron-deficient substrates led to the lowering in yields.

### 2.4. C(sp^2^)—H Oxidation

Bigi and White transformed terminal olefins **141** into α,β-unsaturated ketones **142** via the Wacker oxidation–dehydrogenation process, employing the Pd(II)/PhI(OAc)_2_ co-catalytic system in the presence of 1,4-benzoquinone as an oxidant [77]. Interestingly, PhI(OAc)_2_ **1** played a crucial role as a dehydrogenation catalyst and not as a terminal oxidant. The reaction occurred under mild conditions (35 °C), tolerating a wide range of functional groups, and α, β-unsaturated ketones **142** were obtained in good yields (Figure 44).

Later, similar to Wacker-type oxidation, Fernandes’s research group developed a procedure to convert different aliphatic and aromatic terminal alkenes **143** into functionally diverse methyl ketones **145** using Dess–Martin Periodinane (DMP) **144** as an oxidant under a nitrogen atmosphere [78]. Furthermore, a variety of allylic or homoallylic compounds **146** were examined under similar olefin oxidation conditions to produce substantial quantities of methyl ketones **147**. This approach has several benefits, including excellent functional group compatibility with a wide range of substrates and high yields with complete Markovnikov selectivity (Figure 45).

### 2.5. C—H Phosphorylation/Sulfonation

Huang and colleagues recently demonstrated the Pd(II)-catalyzed sulfonation and phosphorylation of the unactivated benzyl C(sp^3^)—H bonds of 8-methylquinolines **148**, using sulfonate or organophosphorus hypervalent iodine(III) reagents **149** as an oxidant as well as a functional group source [79]. Using this technique, the desirable products **150** or **151** were produced in moderate to high yields over a wide range of substrates (Figure 46). Additionally, the same approach was applicable for the pyridyl-directed C(sp^2^)—H hydroxylation and arylation of arenes.

### 2.6. Miscellenous

In 2015, Kitamura and his research group disclosed a novel route to accessing acyloxyarenes **153** from trimethylsilyl-arenes **152** via a Pd(OAc)_2_-catalyzed desilylative acyloxylation strategy, using the easily available terminal oxidant PhI(OCOCF_3_)_2_ (PIFA) **82** in AcOH [80]. The reaction scope was explored by varying the substituents on arenes as well as by using different carboxylic acids **87** as a source of the acyl group (Figure 47). Additionally, the hydrolysis of acetoxylated products gave access to phenol derivatives, which further extended the synthetic utility of this method. 

A striking example to prepare α-acetoxylated enones **155** from alkynes **154** was developed by Backvall and co-workers via Pd-catalyzed oxidative acetoxylation in the presence of terminal oxidant PhI(OAc)_2_ **1** in DMSO. The addition of 10 mol% of benzoquinone (BQ) elevated the yields of the anticipated products **155 [81]**. The reaction scope was explored by varying the substituents at the propargylic position and also on the arene unit (Figure 48). Further experimental studies using ^18^O-labeled DMSO revealed that the ketonic oxygen atom in the final product originates from dimethylsulfoxide.

Zhu and his co-workers developed a highly efficient and simple route for the synthesis of three types of 2-aminofurans via the Pd-catalyzed cycloisomerization of polysubstituted homoallenyl amides **156**, using hypervalent iodine(III)reagents as the oxidant [82]. The Pd(II)-catalyzed acetoxylative cycloisomerization of **156** was carried out by employing the oxidant PhI(OCOR)_2_ in MeCN under an inert atmosphere to give the desired acetoxylated products **157** in variable yields. When alcohols **118** were used as coupling partners in the presence of oxidant PIFA **82**, the corresponding alkoxylated products **158** were isolated via the alkoxylative cycloisomerization of **156**. Moreover, the hydroxylation of homoallenyl amides **156** under basic conditions furnished the hydroxylated products **159** in significant yields (Figure 49).

## 3. C—C Bond Formation

C—H functionalization using a palladium catalyst is an essential technique in C—C bond-forming reactions. A variety of catalytic reactions involving hypervalent iodine(III) reagents as oxidants have been discussed in this section.

### 3.1. Via Oxidative Cyclization

Li and his colleagues developed an intriguing domino process showcasing the Pd-catalyzed C—H functionalization of N-arylpropiolamides **160**, utilizing iodine(III) reagent **161** as an aryl source [83]. This ingenious procedure resulted in 3-(1-arylmethylene)oxindoles **162** (Figure 50). Furthermore, the study was conducted to find the effect of several electron-rich and electron-deficient substituents on the aryl ring and the terminal triple bond. It was also reported by the group that substrates **160** with the N-acetyl or N-H group were unsuitable for the present reaction. Later, the synthesis of (E)-(2-oxindolin-3-ylidene)phthalimides and (E)-(2-oxoindolin-3-ylidene)methyl acetates by the palladium-catalyzed C—H functionalization of N-arylpropiolamides with phthalimide and carboxylic acids as nucleophiles was also reported by Tang et al. [84,85]. 

Tong et. al. used PIDA **2** as an oxidant to perform the first Pd-catalyzed oxidative cyclization of 1,6-enynes **163** into the corresponding bicyclo [3.1.0] hexane derivatives **165** [86]. Later, Sanford and Tong’s research teams also developed procedures for the synthesis of multi-substituted bicyclo [3.1.0] ring systems by employing bipyridine as a ligand [87,88,89]. Furthermore, Tsujihara et al. reported the enantioselective synthesis of bicyclic lactones **165** from 1,6-enynes **163** in variable yields with up to 95% enantiomeric excess, using asymmetric Pd(II)/Pd(IV) catalysis [90]. The reaction employs chiral ligand spiro bis(isoxazoline) **164** (abbreviated as SPRIXs), a preformed Pd-SPRIX **164** complex as a catalyst, and PhI(OAc)_2_ **1** as a terminal oxidant (Figure 51).

In 2012, the palladium-catalyzed PhI(OAc)_2_-mediated intramolecular trifluoromethylation of alkenes **165** was achieved using TMSCF_3_ **166** as an efficient trifluoromethyl source [91]. This method provided easy-to-access CF_3_-substituted oxindoles **168** at room temperature (Figure 52). The presence of nitrogen-containing ligand **167** and Lewis acid Yb(OTf)_3_ was necessary to obtain the best results for the cyclization reaction. This reaction probably occurs via the arylpalladation of olefins, followed by the nucleophilic attack of arene to generate Pd(II) intermediate **169**, which, upon oxidation and reductive elimination, forms Csp^3—^CF_3_ bond.

Tong and co-authors in 2019 developed an outstanding example of a Pd(II/IV)-catalyzed intramolecular cycloaddition of propargylic alcohol or amine and alkene of substrates **170** through an acetoxylative (3 + 2) annulation approach to afford bicyclic heterocycles **171** in a good yield [92]. A further reaction of enynes **172** in the presence of ligand 1,10-phenanthroline gave cyclopropane products **174** through the formation of 1,10-phenligated Pd(IV) intermediate **173** (Figure 53).

The possible catalytic cycle for this oxidative cycloaddition reaction is given in Figure 54. The reaction starts with acetoxypalladation, which produces alkenyl-Pd(II) intermediate **175**, which, through chair-like transition state (TS) **176**, is then transformed into alkyl-Pd(II) intermediate **177**. Following that, the PhI(OAc)_2_-mediated oxidation yields bicyclic Pd(IV) intermediate **178**, which is then converted to **179** by AcOH loss. Finally, intermediate **179** is subjected to direct C—O reductive elimination to provide product **171** and to renew the palladium catalyst, allowing the catalytic cycle to continue.

### 3.2. Via C—H Bond Arylation

In 2011, Mao and colleagues reported an in-situ Heck-type coupling reaction between olefins **143** and iodobenzene, using hypervalent iodine reagents **180** [93]. The reaction was carried out in an open environment at 40–60 °C using Pd(OAc)_2_ (4 mol%), K_2_CO_3_ as a base, and PEG-400 solvent media. The expected coupling products **181** were obtained in an excellent yield (Figure 55). The catalytic system was devoid of any ligands and had good catalyst recyclability. Magedov and colleagues discovered a comparable Pd-catalyzed Heck-type arylation of terminal alkenes with aryliodine(III) diacetates [94].

Under the optimized reaction conditions, a number of iodobenzene diacetates **183** bearing various functional groups were coupled with benzoxazoles derivatives **182** to efficiently furnish the corresponding arylation products **184** in modest to excellent yields (Figure 56) [95]. A further reaction with iodobenzene instead of PIDA **1** gave a trace amount of the arylation products monitored by GC-MS, which indicates that ArI is not the possible intermediate in the present reaction.

Cai and colleagues used aryliodine(III) diacetates **183** as a coupling partner in the Pd-catalyzed C—H arylation of polyfluoroarenes **185** [96]. The described protocol exhibits excellent substrate scope and tolerates a wide range of functional groups. The reaction mechanism indicates the in situ formation of aryliodides from ArI(OAc)_2_
**183** under basic conditions, resulting in moderate to excellent yields of desirable polyfluorobiaryls **186** (Figure 57).

### 3.3. Via C—H Fluoroalkenylation

Liu et al. published the Pd-catalyzed C—H trifluoromethylation of indoles substituted at C3 position **187** in the presence of terminal oxidant PhI(OAc)_2_ **1**, using the Ruppert–Prakash reagent, TMSCF_3_ **166** [97]**.** The reaction involves the in situ generation of CF_3_ from the reaction of TMSCF_3_ **166** and CsF, which later reacts with indole **187** to afford the desired trifluoromethylated products **188** in variable yields (Figure 58). Meanwhile, the addition of TEMPO and bidentate ligand **167** significantly enhances the product yields. The reaction proposed proceeds via a Pd(II)/Pd(IV) pathway involving the initial electrophillic palladation of a C—H bond of indole to form complex **189**, followed by PIDA-induced oxidation to yield Pd^IV^ intermediate **190**, which finally undergoes reductive elimination to furnish the desired products **188**.

Recently in 2021, Chen et al. reported the Pd-catalyzed chlorodifluoroethylation of various aromatic amides **191** dissolved in DCE with a new 2-chloro,2,2-difluoroethyl(mesityl)iodonium salt (CDFI) **192** in the presence of trifluoroacetic acid (TFA) additive. The reaction was found to be well tolerated in the presence of electron-withdrawing and electron-donating substituents on the aryl ring, affording the chlorodifluoromethylated product **193** in a good to excellent yield. Furthermore, it was observed that adding 4 equiv. of DBU to the DCE solution of the chlorodifluoroethylated substrates obtained from the above reaction resulted in the formation of dehydrochlorination products **194** in high yields at room temperature [98] (Figure 59).

The possible catalytic cycle for chlorodifluoroethylation is given in Figure 60. First, the palladium acetate activation procedure starts the reaction in the presence of trifluoroacetic acid (TFA). The palladium species then builds a cyclometalated intermediate **194** in the presence of the directing group. The complex **194** is then chlorodifluoroethylated by CDFI **192** to give the intermediate **195**. Next, the chlorodifluoroethyl group on the Pd(IV) center is transferred to the aromatic ring to generate intermediate **196**, which finally undergoes the elimination reaction to provide the product **193** and renew the active Pd(II) species for the next catalytic cycle.

Furthermore, in the same year, Novák and coworkers developed Pd-catalyzed ortho C—H activation of the aromatic and heteroaromatic system (Figure 61). Various directing groups (DG), such as the secondary and tertiary amides of anilides, ureas, benzamide derivatives, or ketones, resulted in stereoselective fluorovinylation under mild reaction conditions [99]. The tetrafluoropropenylation reaction of substituted acetanilide **197** with mesityl-(tetrafluoropropenyl) iodonium triflate **198** at 25 °C in the presence of 7.5 mol% palladium(II) acetate catalyst and 2 equiv. trifluoroacetic acid (TFA) dissolved in DCM completed in 4 h to obtain the 2,3,3,3-tetrafluoropropenylated **199** with *Z* selectively in a high yield. The reaction was observed to be tolerant towards various types of substituents.

Besset and coworkers reported an additive-free direct 2,2,2-trifluoroethylation of aryl acrylamides **200** derived from the 8-aminoquinoline as a directing group (DG) by Pd-catalyzed C—H bond activation in the presence of fluorinated hypervalent iodine reagent at room temp. (Figure 62) [100]. The mesityl(trifluoroethyl)iodonium triflate **192**, a hypervalent iodine reagent developed by the Novák group, was used as the coupling partner. The products obtained **201** had a moderate to good yield with stereoselectivity towards the formation of Z-isomers. Under the given reaction conditions, it was observed that the aromatic substitution pattern did not affect the reaction yield. The reaction was well tolerated by a wide range of functional groups, including ester and halogens (Cl, Br, and even I). On changing the directing group, a drastic change in the yield of the products was observed. On using 5-methoxy-8-aminoquinoline as a directing group, the yield of the corresponding product dropped to 18%, while the styrene-directing group resulted in no reaction. Thus, it was concluded that that the directing group played an important role in the transformation.

### 3.4. Via C—H Alkynylation

Waser and colleagues, in 2013, reported the regioselective C2-alkynylation of N-alkylated indoles **202** in the presence of a palladium catalyst and TIPS-EBX **203**, an alkynylating reagent [101], for the first time. The process provided a good yield of 2-alkynylated indoles **204** at room temperature (Figure 63). A variety of substituents, including Cl, Br, F, and I, remained an integral part of the end products, allowing for additional synthetic modifications.

The speculated mechanism for this alkynylation reaction is shown in Figure 64. The reaction initiates by palladation taking place either at the C2 position to form intermediate **206** via concerted metalation–deprotonation (path a) or at the C3 position via electrophilic palladation to provide intermediate **205**, which further undergoes Pd migration to give **206** (path b). Next, the intermediate **206** undergoes oxidative alkynylation using TIPS-EBX **203** to afford Pd(IV) intermediate **207**, which gives the desired product **204** upon undergoing reductive elimination.

### 3.5. Via Coupling

In 2015, Cai and co-workers reported the first example of the Pd-catalyzed homocoupling of aryliodine(III) diacetates **208** towards the synthesis of synthetically useful symmetrical biaryls **209 [102]**. The reaction worked remarkably well under aerobic conditions, required a shorter reaction time, and tolerated a reasonable range of functional groups with good chemoselectivity (Figure 65). Preliminary mechanistic studies revealed in situ generations of aryl iodide through the base-mediated thermal degradation of **208**, accompanied by an Ullmann-type homocoupling to give the desired products **209**.

In 2016, Huang and co-workers illustrated the use of easily available hypervalent iodine(III) compounds **211** as efficient arylating reagents in the preparation of arylated N-heteroaromatic compounds **212** using palladium catalysis [103]. A variety of N-heteroaromatic bromides **210** were successfully coupled to afford aromatic-substituted pyridines and quinolones in moderate to good yields (Figure 66). Furthermore, the substrates with electron-donating groups showed higher reactivity as compared to their counterparts.

The proposed mechanism was studied by taking an example of 3-bromopyridine **210**, and it is shown in Figure 67. The reaction begins with the oxidative addition of **210** with the palladium catalyst to form pyridyl-Pd(II)-Br species **213**, which reacts with the aryl iodide **214** obtained by the thermal degradation of **211** to give intermediate **215**. The reductive elimination of **215** yields coupling product **212** and regenerates the Pd(0) catalyst to continue the catalytic cycle.

Recently, in 2020, Song and co-workers reported a C—H arylation reaction of heterocycle compound **216** in the presence of 5 mol% Pd nanoparticle catalyst **217** and 1.3 equv. hypervalent iodine reagent, [Ph_2_I]BF_4_ **218**, as an oxidant at 60 °C (Figure 68) [104]. The arylation specifically occurred at the C2 position of the heterocyclic compounds, such as indole and furanes, with a high yield of arylated product **119**; only sulphur-containing heterocycles benzothiophene and the substituted thiophene provided C3-arylated products. It was also observed that the yield of the arylated product increased by adding water to the reaction. The Pd nanoparticles used as the catalyst were easily recovered after the reaction and were reutilized five more times. The recycled catalyst provided the arylated product in an 80–86% conversion for up to six cycles.

## 4. C—N Bond Formation

In recent times chemists have shown a lot of interest in catalytic C—H activation/C—N bond formation as it is a robust method for making N-containing aliphatic/aromatic heterocycles. Many methodologies are reported for the C—H aminations of unactivated sp^3^ and sp^2^ C—H bonds employing palladium species as the catalyst and hypervalent iodine reagents as oxidants. In the next section, we will discuss various intra- and intermolecular C(sp^2^/sp^3^)—H bond functionalization reactions reported in the last decade using this strategy.

### 4.1. Via Intramolecular C(sp^2^/sp^3^)—H Bond Functionalization

In 2008, Gaunt and co-workers reported an elegant approach for the synthesis of carbazoles **221** at room temperature [105]. The reaction involves the intramolecular C—H amination of N-substituted biphenyls **220** using Pd(OAc)_2_ and PhI(OAc)_2_ **1** as a catalyst and oxidant, respectively (Figure 69). The possible strategy designed for this reaction involves the coordination of Pd to the amine **220**, cyclopalladation, oxidation, and reductive elimination to afford cyclic products **221**. Further preparation and isolation of the trinuclear carbopalladation complex confirm that the reaction follows the Pd(II)/Pd(IV) catalytic cycle. Furthermore, the potential scope of this method was extended for the synthesis of N-glycosyl carbazoles, a basic skeleton found in many natural products.

Yu and colleagues exhibited that the intramolecular C—H activation/C—H cyclization of phenethylamine derivatives **222** with 2-pyridinesulfonyl as an efficient directing group in the presence of oxidant PhI(OAc)_2_ **1** affords a variety of functional diverse indoline derivatives **223** in moderate to high yields (Figure 70) [106]. Furthermore, 2-pyridinesulfonyl moiety was removed easily under mild conditions by treating with magnesium in MeOH at 0 °C. Similar intramolecular C—H amination reactions to prepare indolines were previously reported by Daugulis and Chen’s research groups independently, using a Pd(OAc)_2_/PhI(OAc)_2_ **1** catalytic system [107,108].

The plausible catalytic cycle for the intramolecular C—H amination of **222** is depicted in Figure 71. Initially, the amine and pyridyl moiety of **222** coordinates with the Pd(II) catalyst to form complex **224**, which undergoes a selective ortho-C—H cleavage to form complex **225**, which, on subsequent oxidation with PhI(OAc)_2_ **1**, gives Pd(IV) species **226**. The reductive elimination of complex **226** forges the desired cyclic products **223** and regenerates the palladium catalyst.

Later, Shi and his research group used 1,2,3-triazoles-4-carboxylic acid as an effective directing group for the palladium-catalyzed C(sp^2^)—H activation of arenes **227** to afford cyclization products **228**, using oxidant PhI(OAc)_2_ **1 [52]**. With both sp^2^ and sp^3^ C—H bonds at the γ-positions of the substrate, the activation occurs selectively at the sp^2^ C—H bond. Moreover, the TAA-directed activation of the methyl sp^3^ C—H bonds of substrates **229** was achieved under the modified condition in the presence of acetic acid to furnish azetidines **230** with a high diastereoselectivity (Figure 72).

Chen’s research group reported the synthesis of heterocyclic amines **232** and **234** by intramolecular C—H amination reactions in toluene, using Pd(OAc)_2_ as catalysts and PhI(OAc)_2_ **1** as an oxidant, under an inert environment [109]. A series of picolinamide (PA)-directed amine substrates **231** and **233** bearing γ- and δ-C(sp^3^)—H bonds were cyclized smoothly to afford azetidines **232** and pyrrolidines **234**, respectively, in significant yields with high diastereoselectivity (Figure 73). Gratifyingly, PA can be removed easily under acidic conditions at room temperature.

### 4.2. Via Intermolecular C(sp^2^)—H Bond Functionalization

Liu and co-workers disclosed the Pd-catalyzed intermolecular oxidative C—H amination of unactivated terminal olefins **235**, using O-alkyl N-sulfonylcarbamates **236** as nitrogen nucleophiles and employing PhI(OPiv)_2_ **56** as a terminal oxidant and 1,4-naphthoquinone as an additive [110]. This oxidative amination protocol leads to the efficient synthesis of valuable allylic amines **237** in useful yields (Figure 74). In addition, the present catalytic system provides products in improved yields as compared to previously reported aerobic oxidative protocols.

Hartwig and colleagues discovered another intriguing approach using the Pd-catalyzed regioselective intermolecular C—H amination of multi-substituted arenes **238** utilizing phthalimide **239** as the source of nitrogen supply [111]. The reactions require the sequential addition of oxidant PhI(OAc)_2_ **1** at 9 and 24 h as it reverts Pd black formed into soluble palladium species, thereby increasing product yields. A series of N-aryl phthalimides **240** were synthesized in moderate to good yields with sterically controlled regioselectivity (Figure 75).

## 5. C—B, C—Si, and C—Halogen Bond Formation

For the first time, Szabó and coworkers reported the selective C—H borylation of simple alkenes, using a palladium pincer complex as an efficient catalyst and PhI(OCOCF_3_)_2_ **82** as an essential oxidant [112]. A series of cyclic **241** and acyclic alkenes **245** were reacted with bis(pinacolato)diboron (B_2_pin_2_) **243** as a boronate source to provide valuable organoboronates **244** and **246**, respectively, in moderate to good yields (Figure 76). Furthermore, the reaction was examined with Pd(OAc)_2_ as a catalyst; however, the products were obtained with lower yields. Except for cycloheptane, which gave allylic compounds preferentially, the subsequent borylation process proceeded with great vinylic selectivity.

Figure 77 depicts the plausible catalytic cycle for the C—H borylation of alkenes **245**. Initially, PhI(OCOCF_3_)_2_ **82** oxidized Pd(II) complex **243** into electrophilic Pd(IV) complex **247**, which underwent further trans-metalation with B_2_pin_2_ **243** to give complex **248**. After that, alkene **242** coordinated with complex **248** to form **249**, which, on subsequent Bpin ligand insertion into the double bond, gave complex **250**. Finally, the reductive elimination–decomplexation of **250** gave targeted products **246** with the regeneration of catalyst **242**.

Szabó and colleagues pioneered the first oxidative allylic C—H silylation of terminal olefins **235** with hexamethyldisilane **251** as the silyl source, yielding allylsilanes **254** [113]. This catalytic process employed hypervalent iodine(III) reagent **253** as an oxidant and a Pd(OAc)_2_ or a nitrogen- and selenium-based palladium catalyst **252** (Figure 78). Moreover, the functional groups, such as ester, benzyl, and amide were well tolerated under the oxidizing conditions, and the anticipated products were obtained with high regio- and stereoselectivity.

The catalytic cycle for the Pd(II)-catalyzed C—H silylation of olefins **235** initiated with the oxidation of a Pd(II) catalyst by **253** to form Pd(IV) complex **255**, followed by the coordination with alkene **235** to give complex **256**, which underwent internal deprotonation to deliver allylpalladium complex **257**. Next, complex **257** underwent transmetalation with hexamethyldisilane **251** to give **258**, which, upon reductive elimination, afforded the final product allylsilanes **254** (Figure 79).

The conversion of a C—H bond into a C—Halogen bond, catalyzed by palladium, is an appealing approach for obtaining valuable aryl halides. However, only a few experiments have been conducted in C—H halogenation reactions using high-valent palladium catalysis in the past. Sanford and colleagues presented the very first study on Pd-catalyzed C—H fluorination using AgF as the fluoride source and PhI(OPiv)_2_ **56** as the oxidant [114]. A variety of 8-methylquinoline analogues **259** with variable substituents were transformed successfully into fluorination products **260** in moderate to good yields (Figure 80). Further substrates **259** bearing electron-withdrawing groups produced better results than those with electron-donating groups.

Rao’s research group demonstrated the ortho C—H iodination of phenol carbamates **261** in DCE/TfOH at room temperature using palladium catalysis and cyclic hypervalent reagent **262** as an iodine source and oxidant [115]. The reaction might follow a Pd(II)/(IV) pathway involving the formation of a cyclopalladium(II) intermediate, oxidation to Pd(IV) intermediate, and a C—I bond reductive elimination to furnish a variety of ortho-iodinated masked phenols **263** with excellent regioselectivity (Figure 81).

## 6. Alkene Difunctionalization

The palladium-catalyzed difunctionalization of simple alkenes using hypervalent iodine reagents has emerged as a powerful method in organic synthesis. Various 1,1- and 1,2-difunctionalization protocols have been developed by several researchers for preparing a diverse array of useful molecules from alkenes. A general mechanism for the Pd-catalyzed oxidative functionalization of alkenes **143** is shown in Figure 82. These transformations often proceed through the formation of δ-alkyl Pd^II^ intermediate **264**, obtained via olefin insertion into the aryl-Pd bond. Such an oxidative Heck intermediate **264** undergoes β-hydride elimination to form Heck product **265**, and the resulting -HPdLn-species readds to give benzylic Pd(II) intermediate **266**, which can be intercepted to furnish 1,1-difunctionalized products **267**. Moreover, Heck intermediates could be oxidatively functionalized into the 1,2-difunctionalized products **268** in the presence of a suitable oxidant.

### 6.1. Pd(II)-Catalyzed 1,1-Difunctionalization of Alkenes

The Pd-catalyzed hypervalent iodine-mediated 1,1-difunctionalization of alkenes is rare, and only a few examples are available in the literature. Moran and co-author published an article highlighting the 1,1-difuntionalization of acrylate derivatives **270** using palladium catalysis [116]. This reaction involves a three-component coupling of substituted arenes **269**, activated alkenes **270**, and hypervalent iodine(III) reagent **271** in acetic acid (Figure 83). The reaction possibly involves the formation of Heck intermediates **264**, which are subsequently functionalized with an acetate ion to give aldol-type products **272**.

Later, Sanford and co-workers disclosed the similar 1,1-aryloxygenation protocol, wherein arylstannanes **273** were successfully coupled with terminal olefins **143** in the presence of hypervalent iodine reagents (PhI(OCOR’)_2_) **180** as an oxidant [117]. This catalytic approach enabled a simultaneous generation of C—C and C—O bonds in a single step, furnishing 1,1-arylacetoxylated products **275** in significant yields (Figure 84). However, the formation of Heck and 1,1-arylchlorinated products was observed under these conditions.

### 6.2. 1,2-Difunctionalization of Alkenes

Significant progress has been made in Pd-catalyzed olefin bis-functionalization, using different hypervalent iodine reagents. Based on this technique, a variety of intra- and intermolecular transformations have also been devised, including diamination, dioxygenation, aminoacetoxylation, fluoroamination, oxidative amination, etc.

#### 6.2.1. Intramolecular 1,2-Difunctionalization of Alkenes

One of the most potent methods for constructing aromatic and aliphatic cyclic compounds containing heteroatom is the catalytic intramolecular difunctionalization of alkenes. The Pd-catalyzed intramolecular oxidative amination for the production of tetrahydrofurans utilizing the PIDA **1** as an oxidant was reported by Sanford and colleagues [118]. Muñiz’s research group then used a palladium catalyst for intramolecular catalytic alkene diamination for the synthesis of bisindoline and cyclic urea scaffolds [119,120,121]. Furthermore, Oshima and co-workers disclosed a novel intramolecular carboacetoxylation protocol for the oxidative cyclization of 4-pentenyl-substituted malonate esters **276**, employing oxidant PhI(OAc)_2_ **1** to afford acetoxymethyl-substituted cyclopentane derivatives **277**, along with bicyclic lactones **278** (Figure 85) [122]. Additionally, the carboacetoxylation products **277** could be easily converted into bicyclic lactones **278** by treating with sulfuric acid under a reflux condition in isopropyl alcohol.

At the same time, Zhu et al. reported the domino carboacetoxylation of N-aryl acrylamides **279** for the synthesis of 3,3′-disubstituted oxindoles **280** in AcOH at 100 °C, using the catalytic quantity of Pd(OAc)_2_ and PhI(OAc)_2_ **1** as an oxidant [123]. Interestingly, when substrate **279** (R^2^ = H) was subjected to domino carboacetoxylation, the expected oxindole was isolated along with spirooxindole **282**. Thus, the authors re-evaluated the present condition and synthesized spirooxindoles **282** from alkenes **281** via a carboamination process under modified conditions, employing the PdCl_2_ catalyst in acetonitrile at 80 °C (Figure 86).

The intramolecular oxyalkynylation of nonactivated terminal alkenes employing hypervalent iodine reagent was reported by Waser and co-workers in 2010 for the first time [124]. Phenol **283** and aliphatic or aromatic acid derivatives **285** in the presence of Pd(hfacac)_2_ as a Pd catalyst and hypervalent iodine(III) reagent derived from benziodoxolone **192** as an acetylene transfer reagent in DCM resulted in a good yield of cyclic ethers **284** and γ-lactones **286**, respectively (Figure 87).

Later, in 2011, the same research group synthesized 4-propargyl lactams **288** by the intramolecular aminoalkynylation of activated olefins **287**, using TIPS-EBX **192** as an alkynylating agent. The catalyst used for the reaction was lithium palladate, Li_2_[PdCl_4_], which was generated in situ [125]. Additionally, the present protocol was successfully utilized for the synthesis of 4-propargyl oxazolidinone and imidazolidinones **290** through the cyclization of allyl carbamates or allyl urea **289** (Figure 88). Furthermore, the synthetic utility of this reaction was extended towards the synthesis of the bicyclic heterocycles pyrrolizidine and indolizidine and also in the total synthesis of the natural product (±)-trachelanthamidine.

Liu and co-workers achieved the Pd-catalyzed intramolecular aminofluorination of unactivated alkenes **291** to fluorine-containing cyclic amines **292** in moderate to high yields [126]. The reaction employed AgF as a fluorinating agent and PhI(OCO^t^Bu)_2_ **56** as a terminal oxidant (Figure 89). These transformations proceeded via a Pd(II/IV) catalytic cycle involving the trans-aminopalladation of olefins mediated by Pd, oxidation by PhI(OPiv)_2_, and a final reductive elimination, giving aminofluorination products.

#### 6.2.2. Intermolecular 1,2-Difunctionalization of Alkenes

Muñiz and colleagues pioneered intermolecular diamination of terminal alkenes in the presence of a Pd catalyst, using saccharin and bissulfonimides as nitrogen donors and iodosobenzene dipivalate **56** as an oxidant [127]. Later, Martinez and Muñiz used a Pd/PhI(OPiv)_2_ catalytic system to effectively perform an intermolecular vicinal diamination of internal alkenes **293** with phthalimide **239** and bissulfonimides **294** as nitrogen sources [128]. The limiting reagent in this reaction was alkene, and the anticipated diamination products **295** were obtained in a variable yield with perfect regio- and diastereoselectivity (Figure 90).

The mechanistic approach towards the intermolecular diamination of alkenes is shown in Figure 91. Initially, alkene **293** coordinates with the Pd catalyst followed by the subsequent aminopalladation involving the nucleophilic addition of **239** trans stereochemistry to form δ-alkylpalladium complex **296**. The rapid oxidation of complex **296** gives Pd(IV) intermediate **297**, which is attacked by bissulfonimides **294** to provide desired products **295** with a net inversion of configuration at the benzylic position.

Similarly, the allylic ethers **298** underwent a catalytic intermolecular 1,2-diamination reaction in the presence of phthalimide **239** and N-fluoro-bis(phenylsulfonyl)imide **299** as nitrogen sources [129]. In the presence of oxidant iodosobenzene dipivalate **56**, the diamination proceeded smoothly with complete regio- and chemoselectivity to furnish the 1,2,3-trisubstituted amination products **300** in good yields (Figure 92).

In continuation, Muñiz and co-workers reported the synthesis of new palladium-phthalimidato complexes **302** and demonstrated their broad applicability as catalysts in the vicinal diamination of allyl ethers **301** and alkenes **235**, using phthalimide **239** and tetrafluorophthalimide as nitrogen sources [130]. The treatment of phthalimide **239** with Pd(OAc)_2_ in nitrile solution at room temperature resulted in the formation of palladium-phthalimidato complexes **302**. The air-stable preformed phthalimidato complexes, which proved to be versatile catalysts for the present diamination reaction, providing the desired products **303** and **304** in useful yields (Figure 93). Furthermore, the same research group synthesized other bissaccharido palladium(II) complexes and investigated their applications in the catalytic regioselective diamination and aminooxygenation of alkenes [131].

The dioxygenation of vicinal alkene is a critical step in the preparation of valuable 1,2-dioxygenated scaffolds. Dong et al. and Shi’s research group simultaneously reported Pd-catalyzed vicinal dioxygenation of olefins using hypervalent iodine reagent as the terminal oxidant through a unique Pd(II)/Pd(IV) mechanism [132,133]. Later, Sanford’s group developed the chiral oxime-directed asymmetric 1,2-dioxygenation of alkenes **305** in the presence of Pd(II) catalysts, using PhI(OBz)_2_ **67** as an oxidant and benzoyloxy source [134]. Various chiral allyl oxime ethers were tested, and the results showed that menthone-derived substrates had the highest reactivity and diastereoselectivity. This method allows the efficient preparation of dibenzoylated compounds **306** with a disterioisomeric ratio up to 90:10. (Figure 94).

Sorensen and Stahl’s groups concurrently reported the Pd-catalyzed aminoacetoxylation of alkenes, using 1 equiv. of nitrogen nucleophiles and 2 equiv. of olefin [135,136]. Muñiz and colleagues later disclosed a modified approach for the intermolecular aminoacetoxylation of internal/terminal alkenes **242** that allowed the alkene substrate to be used as a limiting reagent [137]. Using phthalimide **239** as a nitrogen source, a variety of alkenes, such as allyl ethers, allyl benzenes, (Z)—methylstyrene, etc., were oxidized and transformed into an aminoacetoxylated product **307**. (Figure 95). Based on the results of the experiments, it was shown that PhI(OAc)_2_ **1** alters the stereochemical aspect of the aminoacetoxylation process, favoring the trans-aminopalladation route.

Szabó and colleagues, in 2016, reported the Pd-catalyzed iodofluorination of alkenes **308** using fluoroiodane reagent **309** as an iodine and fluorine source [138]. Pd(BF_4_)_2_(MeCN)_4_ or PdCl_2_(MeCN)_2_ or Pd(OAc)_2_ in CDCl_3_ were used in the reaction. The reaction was planned to proceed to intermediate **311**, which, following C(sp^2^)—I bond breakage, would produce iodofluorinated compounds **310** in moderate to excellent yields (Figure 96). Some alkenes, on the other hand, underwent allylic rearrangement, followed by iodofluorination, to produce internally iodofluorinated compounds. Simple cycloalkenes also produced an iodofluorinated product, but at low yields.

In 2015, Liu and co-workers demonstrated an efficient and simple palladium-catalyzed protocol for the synthesis of β-amino acid derivatives **313** and **315** from alkenes **143** via an intermolecular aminocarbonylation reaction [139]. An array of aliphatic or aromatic terminal alkenes **143** were reacted with either 2-oxazolidone **314** or with phthalimide **239** under a carbon monoxide atmosphere in the presence of a hypervalent iodine(III) reagent as an oxidant (Figure 97). The reaction possessed excellent regioselectivity, broad substrates scope, and remarkable functional group tolerance. Further experimental evidence revealed that the iodine(III) reagent plays a crucial role in accelerating the intermolecular aminopalladation process.

Using PIDA **2** as an oxidant, the same research group devised a new Pd-catalyzed intermolecular oxycarbonylation of terminal **316** or internal alkenes under a CO atmosphere [140]. This difunctionalization procedure allows for the simple synthesis of different oxycarboxylic acids **317** and **319** with high functional group compatibility, regioselectivity, and diastereoselectivity (Figure 98). This method’s potential was expanded to the synthesis of a natural product, (+)-honaucin C, in a 48% yield, up to 99% *ee*.

## 7. Miscellaneous

Das et al., in 2020, introduced an interesting Pd-catalyzed ortho-C(sp^2^)—H variation of (NH)-free 2-substituted benzimidazole, quinazoline, imidazopyridine core as the directing group **320** in the presence of PIDA **1** as a key reagent under mild conditions [141]. Four different functional groups, acetoxy, aryl, iodide, and nitro groups, were installed concurrently on the single substrate by changing the inorganic additives in the presence of PIDA under aerobic conditions. PIDA, a hypervalent iodine catalyst, serves as an oxidant and a source of functional groups in all of the four reactions. In absence of any additive, the acetoxy group becomes attached to the ortho position of the phenyl to give **321**, whereas, in the presence of Cs_2_CO_3_, I_2_, and NaNO_2_, the additives in acetonitrile solvent aryl, iodide, and the nitro group attach at the ortho position to give products **322, 323, 324**, respectively (Figure 99). The reactions complete in 3-6 h and are found to be functional-group tolerant.

## 8. Conclusions

This review explored the various palladium-catalyzed reactions mediated by hypervalent iodine compounds. Hypervalent iodine compounds have emerged as versatile oxidants, with a wide range of reactivity under mild conditions, and at the same time are non-toxic, environmentally friendly, inexpensive, and easy-to-handle reagents in organic synthesis. In recent years, the use of hypervalent iodine reagents in palladium-catalyzed transformations has received a lot of attention as they are strong electrophiles and powerful oxidizing agents. Together, they act as a powerful tool for the diversification of C—H bonds. The intrinsic oxidizing character and specific reactivity with palladium catalysts have successfully synthesized various useful scaffolds through C—O, C—N, C—C, C—Si, C—B, and C—halogen bond formation reactions. In addition, a variety of Pd-catalyzed alkene difunctionalization processes utilizing hypervalent iodine reagents have recently been established. In future, an intriguing area of investigation would be the use of recyclable polymer-supported hypervalent iodine reagents in palladium-catalyzed processes.

## Data Availability

Not applicable.

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
