# Peer review of "Palladium-Catalyzed Organic Reactions Involving Hypervalent Iodine Reagents"

_molecules, 2022, doi:10.3390/molecules27123900_

Round 1

Reviewer 1 Report

Attached

Author Response

Reviewer 1: The response to the comments of reviewer 1 is highlighted in yellow colour in the manuscript.

Comment:  Please remove the “induced” from the title. Instead use Palladium-catalyzed Organic Reactions… or similar

Response: The title is modified as suggested by the reviewer.

Comment: The authors need to check the schemes and numbers more carefully since these do not match in several places (e.g. after Page 25-26, Scheme 40, some of the Scheme numbers are messed up). IN page 29 compound numbers do not match between the discussion and corresponding Schemes.

Response: Modified accordingly

Comment: Line 41-42: Add Stille coupling. It is Sonogashira (not Sonagashira). It is Miyaura (not Miyara)

Response: Modified accordingly (Line 41)

Comment: Line 64: “unactivated arene/alkane C-H bonds”. Did the authors mean arene/alkene?

Response: Both unactivated arene and alkane C-H undergo functionalization (Line 75)

Comment: Scheme 2: Draw the other ligands on Pd(IV) intermediate after oxidative addition of PIDA. In the Scheme 2 sentence, PhI(OAc) should be 2. (not 3).

Response: Modified accordingly (page 3)

Comment: Scheme 4, In the first step when 6 is added to Pd(OPiv)2, pivalic acid (PivOH) should be the side product (not AcOH).

Response: Corrected accordingly (Page 5).

Comment: Line 122: The authors say “valuable intermediate”. Please specify intermediates for what (mention any natural products/drugs for example), which would justify why they are valuable.

Response: The intermediate is involved in various reactions such as Tamao oxidation, Hiyama–Denmark cross-coupling, and nucleophilic addition reaction (Line 136 - 141)

Comment: The authors mention the oxime-masked alcohol as the directing group. It would be better to draw out the oxime in the Scheme 6 instead of just DG. Please also define DG prior to use the acronym.

Response: Instead of DG structure of oxime is drawn in the scheme 6. (page 7)

Comment: Please change the ‘carboxy salt’ to carboxylate salt.

Response: Corrected accordingly.( Line 177)

Comment: Line 184, Please define PIP before using the acronym.

Response: Modified accordingly (Line 201)

Comment: Line 242: …”acetoxylated product 41.” This should be 42.

Response: Corrected accordingly

Comment: Scheme 14: Please write the ligands/Ln on Pd.

Response: Modified accordingly (Page 11)

Comment: Line 269: Rephrase this sentence to avoid redundancy and to make it more clear.

Response: The line is restated as suggested. (Line 281-285)

Comment: “PyrDipSi” – please define and elaborate this before using the abbreviation.

Response: PyrDipSi is defined as pyrimidyldiisopropylsily (Line 292)

Comment: Scheme 17: Is there any regioselectivity reported for this transformation?

Response: Regioselectivity is not reported for the scheme 17 transformation.

Comment: Line 292: …”unactivated sp3 C-H substrates” ,,, there is an extra space in “substrate”

Response: Corrected accordingly (Line 308)

Comment: It would be better to add the thiophene example in the corresponding Scheme.

Response: Thiophene example is added to the scheme 20 (Page 14)

Comment: Scheme 21: Intermediate 62 – draw it completely if it is a dimer. Intermediate 63 – What is X? In general, the authors should be thoroughly elaborate each ligand on Pd when showing the mechanism.

Response: Intermediate 71 is a dimer which is drawn completely and in the  intermediate 72 X denotes the substrate and is mentioned in the scheme 21 (Page 15)

Comment: Line 362: “as as oxidant” … it should be as an oxidant

Response: Corrected accordingly

Comment: Line 366: The authors mention “with good regio- and stereoselectivity”. It would be better to show the values (r.r./d.r./ee) in the Scheme.

Response: The product obtained was reported to be exclusively trans isomer (line 381)

Comment: Line 375: It would be better to show the d.r. in the Scheme or mention it in the  paragraph (either a range or the maximum d.r. obtained).

Response: The value of d.r is added in the text (line 389)

Comment: Scheme 27: Intermediate 80 – Change the Pd-N bond to a solid/dotted bond. Currently it has a hash bond which is used for chiral center. Also draw the full dimer.

Response: Corrected accordingly (page 18)

Comment: Line 409: Fix the error “ page no 10 (Neufeldt and Sanford, 2010)”

Response: Modified accordingly (line 423)

Comment: Scheme 32: It would be better to show the oxidation states of all the intermediates.

Response: Oxidation state of all intermediates are added in scheme 32.(Page 20)

Comment: Line 544: “DG group” – either DG or directing group.

Response: Corrected accordingly

Comment: Line 545-546: It would be better to add a sentence mentioning how these are synthetically useful.

Response: Modified accordingly (Line 576)

Comment: Scheme 40: Intermediate 119 – The oxidation state IV looks like a bond to V. Please fix this.

Response: Scheme 40 is corrected accordingly (Page 25)

Comment: Line 635: Instead of saying “perfect Markovnikov selectivity” it would be better to mention the Markovnikov/anti-Markovnikov ratio in the Scheme/paragraph.

Response: The product obtained in scheme 44 completely follow Markovnikov rule and no other product is obtained. (Line 661)

Comment: Line 677: “Gangguo Zhu and his co-workers” Please change it to Zhu and co-workers

Response: Modified accordingly (Line 703)

Comment: Line 703 – 705 and Line 719-723: Compound numbers are not matching with the corresponding Schemes

Response: Corrected accordingly (Line 741)

Comment: Scheme 62: The CH2CF3 group is missing in the product 193.

Response: Corrected accordingly (Page no 38)

Comment: Scheme 66, 67, Page 38-39: The authors show that the catalyst is PdCl2, which is Pd(II). However, while showing the mechanism the authors say that it starts with Pd(0) species. What is the reducing agent used in this reaction?

Response:  The mechanism is modified. (page 41)

Comment: Line 1011: Please fix the “ (Ye et al., 2013)”

Response: Modified accordingly (Line 1030)

Comment: Scheme 77 on Page 45: Intermediate 239 and 240 and 242. Change the OFOCF3 to OCOCF3.

Response: Corrected accordingly (Page 47)

Reviewer 2 Report

This is an exhaustive and authoritive review of the latest developments in the use of hypervalent iodine(III) compounds in organic synthesis in Pd-catalyzed reactions. The paper is well organized, and the various synthetic procedures are not just only described, but their mechanisms are also treated in detail to explain the rationale for the results. The manuscript is almost completely free of mistakes, except that in line 789 Scheme 54 should be numbered Scheme 56. At some places the captions to the schemes are not on the same page as the scheme itself (e.g. Schemes 24, 39, 55); hopefully this can be arranged properly during production.

Author Response

Reviewer 2 The response to the comments of reviewer 2 is highlighted in blue colour in the manuscript.

Comments: The manuscript is almost completely free of mistakes, except that in line 789 Scheme 54 should be numbered Scheme 56. At some places the captions to the schemes are not on the same page as the scheme itself (e.g. Schemes 24, 39, 55); hopefully this can be arranged properly during production.

Response: Modified accordingly

Reviewer 3 Report

The authors reviewed recent advances in palladium-catalyzed oxidative cross-coupling reactions involving hypervalent iodine reagents, and all reactions were categorized based on the bonds generated, which include C–O, C–C, C–N, C–B, C–Si, and C–halogen bonds, as well as alkene dual-functionalization. There are some significant issues for the authors to consider.

1. Hypervalent iodine reagents are the key roles in this manuscript. A summarized scheme of commonly used iodonium reagents for Pd-catalyzed reactions may be necessary before the main text or every section, which would present a clear overview to readers.

2. The 9th part C(sp2)-H activation in presence of different additives” did not meet the logic of classification in this review. It will be more suitable if this part is moved to Conclusions and Prospect when there is only one research introduced in a section.

3. The content of C-B, C-Si and C-halogen bonds generated reactions only occupied 3-4 pages in a review of more than 60 pages. Please consider to combine the three sections to one.

4. Specific remarks should be given after some abbreviations, e.g. PIDA in Page 3, CMD in Page 6, DCE in Page 10, and HFIP in Page 20.

5. The valency of Pd in schemes should be tagged in one unified form.

6. The authors introduced a paper reported by Wengryniuk's group in 2015. But the cited reference is in 2017. Please check it carefully.

7. The authors evaluated “The catalytic system was devoid of any ligands or additives” in Line 773 Page 32, however PEG-400 was shown in Scheme 55. It is contradictory.

8. The authors should pay much more attentions in details:

In scheme 4: ONE compound was named as 7 and 11, separately.

In scheme 14: The labeling of 39 and 40 are wrong.

ONE compound was named as 99 in scheme 33 and as 93 in scheme 31.

In scheme 40: The valency of Pd in 119 is IV, the V is likely shown in the structure.

There are two 147 in scheme 50 and in scheme 47, separately.

The compounds 155 and 156 in text couldn’t match the structures of 155 and 156 in scheme 49.

In scheme 54: The labels of 158 and 159 are wrong.

There are two schemes of 44 in Page 26 and in page 27.

There are two schemes 54 in Page 32 and in page 33.

There are two schemes 57 in Page 33 and Page 34.

There aren’t serial numbers of all compound in scheme 61.

In scheme 62: ONE compound was named as 192 and 193.

In scheme 21:What’s the meaning of “2” in the structure of compound 62?

In Line 525 Page 21, Pd(OAC)2 should be written as Pd(OAC)2.

In Line 681 Page 28, “PhI(O2COR)2”is a peroxide compound, it may be written wrongly.

In Line 796 Page 33, the compounds 176 in text couldn’t match the structure of 176.

Author Response

Reviewer 3 The response to the comments of reviewer 3 is highlighted in green colour in the manuscript.

Comment 1: Hypervalent iodine reagents are the key roles in this manuscript. A summarized scheme of commonly used iodonium reagents for Pd-catalyzed reactions may be necessary before the main text or every section, which would present a clear overview to readers.

Response 1: As suggested commonly used iodonium reagents for Pd-catalyzed reactions are introduced in main text as figure no 1

 Comment 2: The 9th part C(sp2)-H activation in presence of different additives” did not meet the logic of classification in this review. It will be more suitable if this part is moved to Conclusions and Prospect when there is only one research introduced in a section.

Response 2:  C(sp2)-H activation in presence of different additives is moved to the Miscellaneous section.

Comment 3: The content of C-B, C-Si and C-halogen bonds generated reactions only occupied 3-4 pages in a review of more than 60 pages. Please consider to combine the three sections to one.

Response 3: All 3 sections are combined under sub heading section 5 C-B, C-Si and C-halogen bonds.

Comment 4: Specific remarks should be given after some abbreviations, e.g. PIDA in Page 3, CMD in Page 6, DCE in Page 10, and HFIP in Page 20.

Response 4: All abbreviations are included.

Comment 5: The valency of Pd in schemes should be tagged in one unified form.

Response 5: Modified accordingly.

Comment 6: The authors introduced a paper reported by Wengryniuk's group in 2015. But the cited reference is in 2017. Please check it carefully.

Response 6: Modified accordingly

Comment 7: The authors evaluated “The catalytic system was devoid of any ligands or additives” in

Line 773 Page 32, however PEG-400 was shown in Scheme 55. Response 7:

Response 7: PEG-400 is reported as solvent media.

Comment 8: In scheme 4: ONE compound was named as 7 and 11, separately. The labeling of 39 and 40 are wrong. ONE compound was named as 99 in scheme 33 and as 93 in scheme 31. The valency of Pd in 119 is IV, the V is likely shown in the structure. There are two 147 in scheme 50 and in scheme 47, separately. The compounds 155 and 156 in text couldn’t match the structures of 155 and 156 in scheme 49. There are two schemes of 44 in Page 26 and in page 27. There are two schemes 54 in Page 32 and in page 33. There are two schemes 57 in Page 33 and Page 34. There aren’t serial numbers of all compound in scheme 61. In scheme 62: ONE compound was named as 192 and 193. What’s the meaning of “2” in the structure of compound 62? In Line 525 Page 21, Pd(OAC)2 should be written as Pd(OAc)2. In Line 681 Page 28, “PhI(O2COR)2”is a peroxide compound, it may be written wrongly. In Line 796 Page 33, the compounds 176 in text couldn’t match the structure of 176.

Response 8: Modified accordingly and highlighted in green colour

Reviewer 4 Report

In this review the authors explored various palladium-catalyzed reactions mediated by hypervalent iodine compounds which includes C–O, C–N, C–C, C–Si, C–B and C–halogen bond formation to produce wide varieties of compounds. Overall, after addressing the points mentioned below, I recommend this article to publish in Molecules.

  1. The article composed of concise information about hypervalent iodine mediated cross coupling reaction using palladium chemistry. The authors have almost covered all types of reactions with trivial examples in this specified area including their own publications.
  2. In introduction section ample number of references were presented and it will be more interesting if the authors show the types of hypervalent iodine reagents described as well as the general reaction mechanism for any transformation mediated by the hypervalent iodine compounds. Further, it is recommended to cite following relevant articles along with ref: 10-15 related to importance of hypervalent iodine reagents in carbohydrate chemistry. i) Chennaiah, A.; Vankar, Y. D. One-Step TEMPO-Catalyzed and Water-Mediated Stereoselective Conversion of Glycals into 2-Azido2-deoxysugars with a PIFA-Trimethylsilyl Azide Reagent System. Org. Lett. 2018, 20, 2611−2614 ii) Chennaiah, A.; Verma, A. K.; Vankar, Y. D. TEMPO-Catalyzed Oxidation of 3-O-Benzylated/Silylated Glycals to the Corresponding Enones Using a PIFA−Water Reagent System. J. Org. Chem. 2018, 83, 10535−10540 iii) Chennaiah, A.; Bhowmick, S.; Vankar, Y. D. Conversion of glycals into vicinal-1,2- diazides and 1,2-(or 2,1)-azidoacetates using hypervalent iodine reagents and Me3SiN3. Application in the synthesis of N-glycopeptides, pseudo-trisaccharides and an iminosugar. RSC Adv. 2017, 7, 41755−41762.
  3. The types of reactions are categorized systematically in this article based on the different C-X bond formations. The reactions schemes were drawn properly with correct references and elaborated flawlessly.
  4. The reaction mechanisms were presented correctly in line with the original references.
  5. The authors can add a recent literature in the section “Misc” about alkoxylation of inactivated C(sp3)–H bonds catalyzed by palladium(ii) complexes. (Chem. Sci., 2021, 12, 7185–7195).
  6. The authors have already published a review article titled as “Hypervalent Iodine Reagents in Palladium-Catalyzed Oxidative Cross-Coupling Reactions” in the journal Frontiers in Chemistry in 2020 (https://doi.org/10.3389/fchem.2020.00705). This review article also covers almost the same types of reactions presented in the current manuscript. In addition, the authors added some of the recent reports in present review and since it is a very specific area, (hypervalent iodine chemistry) within a short span of time (from 2020) naturally very few articles are published.

Author Response

Reviewer 4 Response to the comments of reviewer 4 is highlighted in purple colour in the manuscript.

Comment 1:  In the introduction section ample number of references were presented and it will be more interesting if the authors show the types of hypervalent iodine reagents described as well as the general reaction mechanism for any transformation mediated by the hypervalent iodine compounds.

Response 1: Examples of hypervalent iodine (III)/(V) reagents are included in the introduction section in Figure 1.

Comment 2: Further, it is recommended to cite following relevant articles along with ref: 10-15 related to importance of hypervalent iodine reagents in carbohydrate chemistry. i) Chennaiah, A.; Vankar, Y. D. One-Step TEMPOCatalyzed and Water-Mediated Stereoselective Conversion of Glycals into 2-Azido2- deoxysugars with a PIFA-Trimethylsilyl Azide Reagent System. Org. Lett. 2018, 20, 2611−2614

ii) Chennaiah, A.; Verma, A. K.; Vankar, Y. D. TEMPO-Catalyzed Oxidation of 3-O-Benzylated/Silylated Glycals to the Corresponding Enones Using a PIFA−Water Reagent System. J. Org. Chem. 2018, 83, 10535−10540 iii) Chennaiah, A.; Bhowmick, S.; Vankar, Y. D. Conversion of glycals into vicinal-1,2- diazides and 1,2-(or 2,1)-azidoacetates using hypervalent iodine reagents and Me3SiN3. Application in the synthesis of N-glycopeptides, pseudo-trisaccharides and an

iminosugar. RSC Adv. 2017, 7, 41755−41762.

Response 2: All the suggested articles are sited as reference no 16, 17, 18. Highlighted in purple colour.

Comment 3: The authors can add recent literature in the section “Misc” about alkoxylation of inactivated C(sp3)–H bonds catalyzed by palladium(ii) complexes. (Chem. Sci., 2021, 12, 7185–7195).

Response 3: The literature is added in 2.2.2 C(sp3)‒H acyloxylation section page no 22.

Round 2

Reviewer 1 Report

Accept in present form